# Dense Retrieval as Indirect Supervision for Large-space Decision Making

**Nan Xu**🐱 **Fei Wang**🐱 **Mingtao Dong**🐱 **Muhao Chen**🐹

🐱University of Southern California 🐹University of California, Davis
{nanx,fwang598,mingtaod}@usc.edu muhchen@ucdavis.edu

## Abstract

Many discriminative natural language understanding (NLU) tasks have large label spaces. Learning such a process of large-space decision making is particularly challenging due to the lack of training instances per label and the difficulty of selection among many fine-grained labels. Inspired by *dense retrieval* methods for passage finding in open-domain QA, we propose a reformulation of large-space discriminative NLU tasks as a learning-to-retrieve task, leading to a novel solution named **D**ense **D**ecision **R**etrieval (*DDR* 🐱). Instead of predicting fine-grained decisions as logits, *DDR* adopts a dual-encoder architecture that learns to predict by retrieving from a decision thesaurus. This approach not only leverages rich indirect supervision signals from easy-to-consume learning resources for dense retrieval, it also leads to enhanced prediction generalizability with a semantically meaningful representation of the large decision space. When evaluated on tasks with decision spaces ranging from hundreds to hundred-thousand scales, *DDR* outperforms strong baselines greatly by $27.54\%$ in $P@1$ on two extreme multi-label classification tasks, $1.17\%$ in F1 score ultra-fine entity typing, and $1.26\%$ in accuracy on three few-shot intent classification tasks on average.[1]

## 1 Introduction

Many discriminative natural language understanding (NLU) tasks require making fine-grained decisions from a large candidate decision space. For example, a task-oriented dialogue system, when responding to users' requests, needs to frequently detect their intents from hundreds of options (Zhang et al., 2020b; Ham et al., 2020). Description-based recommendation in e-commerce needs to search from millions of products in response to users' descriptions (Gupta et al., 2021; Xiong et al., 2022).

Previous studies for such NLU tasks still train classifiers as the solution (Gupta et al., 2021; Yu et al., 2022; Lin et al., 2023a). However, we argue that this straightforward approach is less practical for large-space decision making for several reasons. First, more decision labels naturally lead to data scarcity, since collecting sufficient training data for all labels will need significantly more cost. This issue is also accompanied by the issue of rare labels in the long tail of highly skewed distributions suffering severely from the lack of sufficient training instances (Zhang et al., 2023), leading to overgeneralization bias where frequent labels are more likely to be predicted compared with rare ones (Xu et al., 2022). Second, the non-semantic logit-based representation for decision labels in a classifier also makes the model hard to generalize to rarely seen labels, and not adaptable to unseen ones in training. This issue also impairs the applicability of the decision-making model to real-world scenarios, such as recommendation (Xiong et al., 2022) and task-oriented parsing (Zhao et al., 2022a), where the decision space may expand rapidly and crucial labels may be absent in training.

On the contrary, motivated by semantic similarity of examples annotated with identical labels, recent studies propose contrastive learning schemes that leverage Siamese encoding architectures to maximize similarity scores of representation for positive example pairs (Henderson et al., 2020; Zhang et al., 2020a; Dahiya et al., 2021a; Mehri and Eric, 2021; Zhang et al., 2021; Xiong et al., 2022). Meanwhile, inductive bias from NLU tasks such as masked language modeling (Mehri et al., 2020; Dai et al., 2021) and natural language inference (NLI; Li et al. 2022; Du et al. 2022) has shown beneficial for learning on rare and unseen labels via indirect supervision (Yin et al., 2023). However, when dealing with very large decision spaces, existing methods still face critical trade-offs between

---

[1]Code and resources are available at https://github.com/luka-group/DDR.

generalizability and efficiency of prediction.[2]

Inspired by the recent success of dense retrieval methods that learn to select answer-descriptive passages from millions of candidate documents for open-domain QA (Karpukhin et al., 2020; Lee et al., 2021; Zhan et al., 2021), we propose an indirectly supervised solution named **D**ense **D**ecision **R**etrieval (*DDR* 🧭). *DDR* provides a general reformulation of large-space decision making as learning to retrieve from a semantically meaningful decision thesaurus constructed based on task-relevant resources. The model adopts the dual-encoder architecture from dense retrieval models to embed input texts and label descriptions, and learns to predict by retrieving from the informative decision thesaurus instead of predicting fine-grained decisions as logits. In this way, *DDR* not only leverages rich indirect supervision signals from other easy-to-consume learning resources for dense retrieval. It also leads to enhanced prediction performance and generalizability with a semantically meaningful representation of the large decision space. We evaluate *DDR* on large decision spaces ranging from hundreds for *few-shot intent detection*, ten-thousand for *ultra-fine entity typing*, to hundred-thousand scales for *extreme multi-label classification*. *DDR* obtains state-of-the-art performance on 6 benchmark datasets in multiple few-shot settings, improving the most competitive baselines by $27.54\%$ in P@1 for extreme classification, $1.17\%$ in F1 for entity typing and $1.26\%$ in accuracy for intent detection on average. Ablation studies show that both the constructed informative label thesaurus and indirect supervision from dense retrieval contribute to performance gain.

The technical contributions of this work are three-fold. First, we present a novel and strong solution, *DDR*, for NLU tasks with large-space decision making that leverages indirect supervision from dense retrieval. Second, we provide semantically meaningful decision thesaurus construction that further improves the decision-making ability of *DDR*. Third, we comprehensively verify the effectiveness of *DDR* on tasks of fine-grained text classification, semantic typing and intent detection where the size of decision spaces range from hundreds to hundreds of thousands.

---

[2]For example, NLI-based inference leads to $k$ times more inference cost where $k$ being the size of the decision space, which is prohibitive for large-space decision making tasks.

## 2 Related Work

**Indirect Supervision** Indirectly supervised methods (Roth, 2017; He et al., 2021; Yin et al., 2023) seek to transfer supervision signals from a more resource-rich task to enhance a specific more resource-limited task. A method of this kind often involves reformulation of the target task to the source task. Previous studies have investigated using source tasks such as NLI (Li et al., 2022; Yin et al., 2019; Lyu et al., 2021, inter alia), extractive QA (Wu et al., 2020; FitzGerald et al., 2018; Li et al., 2020, inter alia), abstractive QA (Zhao et al., 2022a; Du and Ji, 2022) and conditioned generation (Lu et al., 2022; Huang et al., 2022b; Hsu et al., 2022, inter alia) to enhance more expensive information extraction or semantic parsing tasks. Recent studies also transformed these technologies to specialized domains such as medicine (Xu et al., 2023) and software engineering (Zhao et al., 2022b) where model generalization and lack of annotations are more significant challenges. There has also been foundational work which studies the informativeness of supervision signals in such settings (He et al., 2021).

However, the aforementioned studies are not designed for discriminative tasks with very large decision spaces, and do not apply directly due to issues such as high inference costs (NLI) and requiring decisions to be inclusive to the input (QA). We instead propose dense retriever that naturally serves as a proper and efficient form of indirect supervision for large-space decision making.

**NLU with Large Decision Spaces** Many concrete NLU tasks deal with large decision spaces, including description-based recommendation in e-commerce (Gupta et al., 2021; Xiong et al., 2022) and Web search (Gupta et al., 2021), user intent detection in task-oriented dialog systems (Zhang et al., 2020b; Ham et al., 2020), and fine-grained semantic typing (Choi et al., 2018; Chen et al., 2020), etc. Previous studies for such tasks either train classifiers (Yu et al., 2022; Lin et al., 2023a) or rely on contrastive learning from scratch (Zhang et al., 2021; Xiong et al., 2022), which are generally impaired by insufficiency of training data and hard to generalize to rarely seen and unseen labels. These challenges motivate us to explore a practical solution with indirect supervision from a dense retriever.

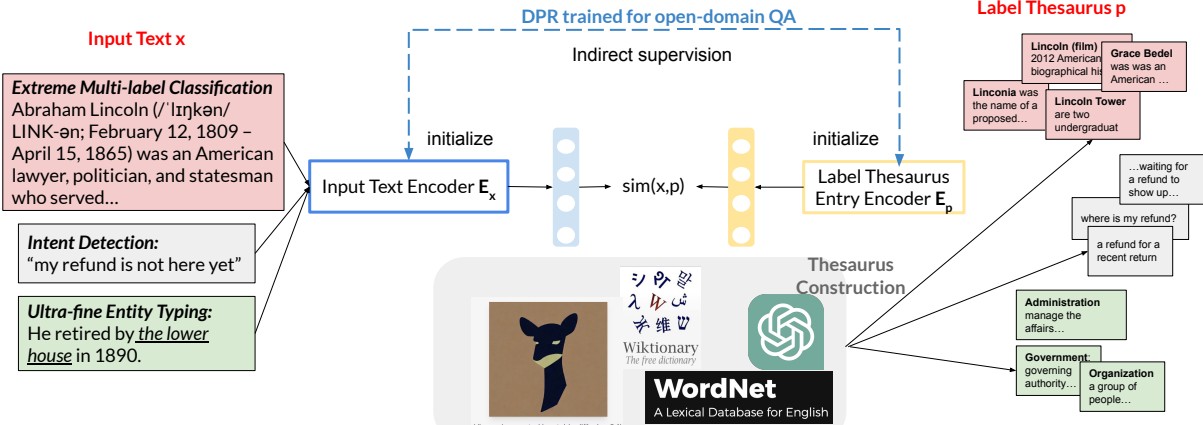

Figure 1: Overview of *DDR* that reformulates general large-space decision making tasks as dense retrieval. Label thesaurus is constructed with detailed descriptions from publicly available resources and the dual-encoder learns to maximize similarity between embeddings of input and label thesaurus entry with indirect supervision from a pre-trained dense retriever.

## 3 Method

We first describe the reformulation of large-space decision making task as dense retrieval (§3.1). We then introduce several automatic ways to construct label thesaurus of high quality (§3.2). Lastly, we demonstrate the dual-encoder architecture of *DDR* (illustrated in Fig. 1) to retrieve decisions (§3.3).

### 3.1 Preliminary

The mechanism of dense retrieval for general decision-making in large spaces can be formalized as follows. Given a textual input, such as "Illy Ground Espresso Classico Coffee: Classico, classic roast coffee has a lingering sweetness and delicate..." for description-based product recommendation, a model learns to infer their corresponding labels by mapping input to textual descriptions in a large label thesaurus (such as names, brands, component, etc.), instead of simply predicting the label index.

Formally speaking, given a textual input $x$, a decision space covering $L$ labels and a label thesaurus that contains corresponding entries (label descriptions) $D = \{d_1, d_2, \ldots, d_L\}$. We first split every entry into text passages of a maximum length (e.g., 100 words) as the basic retrieval units and get $M$ total passages $P = \{p_1, p_2, \ldots, p_M\}$[3]. A retriever $R : (x, P) \to P_R$ is a function that takes as input a text sequence $x$ and the label thesaurus $P$ and

returns a ranking list of passages $P_R$, where labels represented by higher ranked passages are more likely to be relevant to the input. As demonstrated in Fig. 1, *DDR* also leverages indirect supervision from open-domain QA by initializing parameters of retriever $R$ with those from DPR.

### 3.2 Thesaurus Construction

Depending on the target task, the decision thesaurus can provide knowledge about the decision space from various perspectives. Given label names, we create informative descriptions for each label automatically without extra expenses. We can refer to publicly available resources for constructing thesaurus of high quality. To name a few:

*1) Lexicographical knowledge bases*: dictionaries such as WordNet (Miller, 1995) and Wiktionary[4] provide accurate definitions as well as plentiful use cases to help understand words and phrases.

*2) Large Language Models*: LLMs trained on instructions like ChatGPT and Vicuna-13B (Chiang et al., 2023) are able to generate comprehensive and reasonable descriptions of terms when they are prompted for explanation.

*3) Training examples*: directly adopting input text when labels with detailed information also appear as example input, or aggregating input texts from multiple examples assigned with the same label if they share very high-level semantic similarities.

We describe thesaurus construction in practice for each investigated downstream task in §4.

---

[3]Since a label-descriptive entry, such as those in XMC tasks, is sometimes too long to encode with a Transformer encoder, we break it down into multiple passages to avoid the quadratic dependency w.r.t input length.

[4]https://en.wiktionary.org/

### 3.3 Learning to Retrieve Decisions

**Encoders** Since a key motivation of *DDR* is to leverage indirect supervision from retrieval tasks, we adopt the dual-encoder architecture similar to a dense retriever such as DPR (Karpukhin et al., 2020), where two sentence encoders with special tokens (BERT (Devlin et al., 2019) by default), $E_x$ and $E_p$, are leveraged to represent input text $x$ and passage $p$ individually.[5] The similarity between input and passage is computed using dot product of their vectors denoted by $\text{sim}(x, p)$.

**Training** The objective of the reformulated large-space decision making problem is to optimize the dual encoders such that relevant pairs of input text and label description (thesaurus entry) should have higher similarity than irrelevant pairs in the embedding space. Hence, the overall learning objective is a process of contrastive learning.

To tackle single-label classification where each instance contains one input $x_i$, one relevant (positive) passage $x_i^+$, along with $n$ irrelevant (negative) passages $p_{i,j}^-$, we minimize the negative log likelihood of the positive label description

$$L(x_i, p_i^+, p_{i,1}^-, \ldots, p_{i,n}^-) \tag{1}$$
$$= -\log \frac{e^{\text{sim}(x_i, p_i^+)}}{e^{\text{sim}(x_i, p_i^+)} + \sum_{j=1}^{n} e^{\text{sim}(x_i, p_{i,j}^-)}}.$$

We extend to multi-label classification tasks where each input has multiple positive passages and more than one of them participates in model update simultaneously. Accordingly, the cross entropy loss is minimized to encourage similarity between $m$ positive pairs instead:

$$L(x_i, p_{i,1}^+, \ldots, p_{i,m}^+, p_{i,1}^-, \ldots, p_{i,n}^-) \tag{2}$$
$$= -\sum_{k=1}^{m} \log \frac{e^{\text{sim}(x_i, p_{i,k}^+)}}{\sum_{j=1}^{m} e^{\text{sim}(x_i, p_{i,j}^+)} + \sum_{j=1}^{n} e^{\text{sim}(x_i, p_{i,j}^-)}}.$$

*Positive, negative and hard negative passages:* the decision thesaurus entries that describe true labels in thesaurus for each input are deemed as positive. In-batch negatives are positive entries for other input in the same mini-batch, making the computation more efficient while achieving better performance compared with random passages or other negative entries (Gillick et al., 2019;

---

 If not otherwise specified, we train with positive and in-batch negatives to obtain a weak *DDR* model firstly; we then retrieve label thesaurus with the weak model for training examples and leverage wrongly predicted label description as hard negatives, which are augmented to training data to obtain a stronger *DDR*.

We note that, different from open-domain QA where one positive pair can be considered per time with Eq. 1, considering multiple positive pairs is very important for tasks where the decision space though large but follows highly skewed distributions, when in-batch negatives are adopted for model update. In the mini-batch, multiple inputs may share same labels that are popular in the head of long-tail label distribution. In this case, these input instances have multiple positive passages: one randomly sampled from label passages as normal and some from positives of other in-batch inputs.

**Inference** After training is completed, we encode all passages in label thesaurus once with $E_p$ and index them using FAISS offlne (Johnson et al., 2019). Given a new input, we obtain its embedding with $E_x$ and retrieve the top-$k$ passages with embeddings closest to that of input.

## 4 Experiments

We evaluate *DDR* on three NLU tasks with large deicsion spaces in two criteria, 1) a minority of label space is observed during training: *Extreme Multi-label Classification* (§4.1) and *Entity Typing* (§4.2), or 2) limited amounts of examples are available per label: *Intent Classification* (§4.3). We investigate decision spaces of up to hundreds of thousands. Dataset statistics are shown in Appx. Tab. 1.

### 4.1 Extreme Multi-label Classification

**Task** Many real-world applications can be formulated as eXtreme Multi-label Classification (XMC) problems, where relevant labels from the set of an extremely large size are expected to be predicted given text input. Considering the ever-growing label set with newly added products or websites and the time-consuming example-label pairs collection, we follow the few-shot XMC setting (Gupta et al., 2021; Xiong et al., 2022) to choose relevant labels from all available seen and unseen labels.[6]

---

[5]For example, in cases where a BERT model is used, the [CLS] token is used for representation of input text and the decision thesaurus entries

[6]By randomly sampling 1% and 5% labels from the large decision space, we only use their instances for model training and predict on the test set covering the whole label space.

| Dataset | Setting | Train | Dev | Test | Labels |
|---|---|---|---|---|---|
| **Extreme Multi-label Classification** | | | | | |
| LF-Amazon-131K | 1% | 6227 | - | 134,835 | 131,073 |
| | 5% | 29,719 | - | | |
| LF-WikiSeeAlso-320K | 1% | 13,920 | - | 177,515 | 312,330 |
| | 5% | 62,704 | - | | |
| **Ultra-fine Entity Typing** | | | | | |
| UFET | Crowd | 1998 | 1998 | 1998 | 10,331 |
| **Intent Classification** | | | | | |
| BANKING | 5-shot | 385 | 1540 | 3080 | 77 |
| | 10-shot | 770 | | | |
| HWU | 5-shot | 320 | 1076 | 1076 | 64 |
| | 10-shot | 640 | | | |
| CLINC | 5-shot | 750 | 3000 | 4500 | 150 |
| | 10-shot | 1500 | | | |

Table 1: Statistics and experimental settings on datasets with label spaces ranging from 64 to 320K.

**Datasets and Metrics** The public benchmark dataset LF-Amazon-131K collects pairs of relevant products with textual descriptions for commercial recommendation, while LF-WikiSeeAlso-320K[7] aims to link relevant Wikipedia passages for reference (Bhatia et al., 2016; Gupta et al., 2021).

We use metrics widely adopted by XMC (Chien et al., 2023) and IR (Thakur et al., 2021) literature: precision@k for the proportion of the top-$k$ predicted labels to be true labels, and recall@k for the proportion of true labels found in the top-$k$ predictions.[8]

**Baselines** We compare *DDR* with strong XMC baselines in three categories: 1) Transformer-based models that learn semantically meaningful sentence embeddings with siamese or triplet network structures, including Sentence-BERT (Reimers and Gurevych, 2019) and MPNet (Song et al., 2020); 2) competitive methods originally proposed for scalable and accurate predictions over extremely large label spaces, including XR-Linear (Yu et al., 2022) that recursively learns to traverse an input from the root of a hierarchical label tree to a few leaf node clusters, Astec (Dahiya et al., 2021b) with four sub-tasks for varying trade-offs between accuracy

and scalability, and SiameseXML (Dahiya et al., 2021a) based on a novel probabilistic model to meld Siamese architectures with high-capacity extreme classifiers; 3) methods specifically designed to improve performance with few-shot labels available, including ZestXML (Gupta et al., 2021) that learns to project a data point's features close to the features of its relevant labels through a highly sparsified linear transform, and MACLR (Xiong et al., 2022) that pre-trains Transformer-based encoders with a self-supervised contrastive loss.

**Implementation Details** The label space of LF-WikiSeeAlso-320K is composed of relevant passage titles, hence the corresponding full content should be available in Wikipedia and we use them as the label thesaurus.[9] For relevant product advertisement and recommendation, concrete product description provides sufficient information as the label thesaurus for LF-Amazon-131K. We extract product textual descriptions mainly from Amazon data in XMC benchmark (Bhatia et al., 2016) and public Amazon product meta datasets (Ni et al., 2019), leading to 95.67% of labels well documented. As introduced in §3.2, we then use Chat-GPT [10] to obtain relevant information for remaining undocumented labels [11].

We perform label thesaurus construction on Intel(R) Xeon(R) Gold 5217 CPU @ 3.00GHz with 32 CPUs and 8 cores per CPU. It takes 59.12 minutes to construct label thesaurus for LF-WikiSeeAlso-320K, while 28.24 minutes for LF-Amazon-131K. We leverage the same infrastructure to prepare label thesaurus in following tasks unless otherwise specified.

Due to a lack of explicit knowledge about the connection between the majority of labels and their examples for training, *DDR* only learns to distinguish positive example-label pairs from in-batch negatives without further negative mining. For inference, with the constructed label thesaurus, we first retrieve the large label space with a test example based on Sentence-BERT embeddings of the title and the first sentence in an unsupervised way. After training *DDR* with few-shot labels, we mimic the practice of DPR+BM25 for open-

---

[7]DPR was trained on the clean, text-portion of articles from the Wikipedia dump for question answering. Lists such as the "See also" section have been removed during pre-processing. Hence example-label pairs in LF-WikiSeeAlso-320K are unseen during DPR training and it is fair to fine-tune DPR on this task.

[8]Following (Xiong et al., 2022), we use $k = \{1, 3, 5\}$ for precision@k and $k = \{1, 3, 5, 10, 100\}$ for recall@k.

[9]Since the specific dump version is unclear, we look for different resources (e.g., latest dump, train and test passages, Wikipedia API search) and cover 99.95% of labels in the end.

[10]https://chat.openai.com/

[11]Similarly, for following tasks, we obtain label information by prompting LLMs if the label definition or description is unavailable from existing resources.

| Method | 1% Labels | | | | | | | | 5% Labels | | | | | | | |
| | Precision | | | Recall | | | | | Precision | | | Recall | | | | |
| | @1 | @3 | @5 | @1 | @3 | @5 | @10 | @100 | @1 | @3 | @5 | @1 | @3 | @5 | @10 | @100 |
| --- | --- | --- | --- | --- | --- | --- | --- | --- | --- | --- | --- | --- | --- | --- | --- | --- |
| **LF-Amazon-131K** | | | | | | | | | | | | | | | | |
| XR-Linear | 1.53 | 0.57 | 0.36 | 0.67 | 0.75 | 0.78 | 0.81 | 0.92 | 5.09 | 2.09 | 1.32 | 2.36 | 2.86 | 3.02 | 3.18 | 3.74 |
| Astec | 0.94 | 0.44 | 0.29 | 0.55 | 0.78 | 0.84 | 0.91 | 1.13 | 3.94 | 1.92 | 1.26 | 2.31 | 2.34 | 3.66 | 4.00 | 4.96 |
| SiameseXML | 1.45 | 0.56 | 0.35 | 0.84 | 0.96 | 1.00 | 1.03 | 1.16 | 5.36 | 2.23 | 1.41 | 3.15 | 3.89 | 4.08 | 4.27 | 4.82 |
| ZestXML | 10.10 | 9.19 | 7.34 | 5.63 | 14.46 | 18.61 | 23.73 | 32.69 | 12.33 | 10.00 | 8.71 | 6.84 | 17.19 | 21.97 | 28.19 | 46.49 |
| Sentence-BERT | 12.64 | 9.82 | 7.80 | 6.97 | 15.34 | 19.74 | 25.33 | 43.53 | 15.47 | 12.24 | 9.64 | 8.63 | 19.23 | 24.40 | 30.82 | 49.22 |
| MPNet | 14.78 | 11.55 | 8.97 | 8.28 | 18.24 | 22.84 | 28.54 | 45.89 | 15.03 | 11.88 | 9.28 | 8.47 | 18.74 | 23.69 | 29.93 | 48.84 |
| MACLR | 18.74 | 16.07 | **12.52** | 10.73 | 25.44 | **31.89** | **39.17** | 57.55 | 19.56 | 16.19 | 12.64 | 11.15 | 25.65 | 32.18 | 39.63 | 58.45 |
| *DDR* (w/o thesaurus) | 15.39 | 10.92 | 7.87 | 8.25 | 19.15 | 24.10 | 31.51 | 54.32 | 21.21 | 13.77 | 10.23 | 12.02 | 24.18 | 30.29 | 38.19 | 58.75 |
| *DDR* (scratch) | 19.55 | 14.48 | 11.50 | 10.87 | 22.50 | 29.00 | 37.80 | 56.53 | 23.77 | 16.48 | 12.75 | 13.31 | 25.54 | 31.85 | 40.46 | 60.30 |
| *DDR* (full) | **22.39** | **16.38** | 12.40 | **12.57** | **25.64** | 31.44 | 38.44 | **57.72** | **25.22** | **17.96** | **13.63** | **14.18** | **27.94** | **34.40** | **41.72** | **61.17** |
| **LF-WikiSeeAlso-320K** | | | | | | | | | | | | | | | | |
| XR-Linear | 1.24 | 0.57 | 0.37 | 0.42 | 0.58 | 0.63 | 0.68 | 0.76 | 4.69 | 2.20 | 1.46 | 1.82 | 2.41 | 2.63 | 2.82 | 3.42 |
| Astec | 1.25 | 0.60 | 0.41 | 0.69 | 0.98 | 1.11 | 1.27 | 1.56 | 5.90 | 2.80 | 1.86 | 3.26 | 4.49 | 4.95 | 5.49 | 6.83 |
| SiameseXML | 1.81 | 0.75 | 0.48 | 1.03 | 1.26 | 1.33 | 1.41 | 1.67 | 6.83 | 3.15 | 2.06 | 3.88 | 5.15 | 5.56 | 6.02 | 7.09 |
| ZestXML | 8.74 | 6.78 | 5.41 | 4.68 | 9.70 | 12.21 | 15.73 | 24.98 | 10.06 | 8.11 | 6.60 | 5.33 | 11.49 | 14.74 | 19.57 | 40.46 |
| Sentence-BERT | 16.30 | 12.62 | 10.08 | 9.30 | 18.92 | 23.78 | 30.40 | 52.92 | 18.47 | 14.19 | 11.29 | 10.82 | 21.55 | 26.77 | 33.92 | 57.02 |
| MPNet | 17.14 | 12.64 | 9.96 | 9.98 | 18.98 | 23.45 | 29.67 | 50.75 | 18.59 | 13.99 | 11.08 | 10.89 | 21.12 | 26.10 | 32.82 | 54.70 |
| MACLR | 19.09 | 14.57 | 11.53 | 11.39 | 22.34 | 27.63 | 34.81 | 57.92 | 20.99 | 15.57 | 12.26 | 12.59 | 23.94 | 29.41 | 36.78 | 59.81 |
| *DDR* (w/o thesaurus) | 17.52 | 15.93 | 14.23 | 4.89 | 13.11 | 18.88 | 25.93 | 46.93 | 18.14 | 16.65 | 15.39 | 5.12 | 14.10 | 21.81 | 30.00 | 52.54 |
| *DDR* (scratch) | 21.26 | 15.81 | 13.08 | 11.16 | 22.52 | 29.55 | 40.14 | 64.94 | 25.04 | 18.95 | 15.05 | 12.46 | 25.70 | 32.18 | 40.24 | 66.22 |
| *DDR* (full) | **24.70** | **18.45** | **14.69** | **13.04** | **26.31** | **32.99** | **41.08** | **65.39** | **27.78** | **19.98** | **15.66** | **15.03** | **28.87** | **35.47** | **43.35** | **68.66** |

Table 2: Results of few-shot XMC where the training subset covers 1% (left) and 5% (right) labels from the whole set. *DDR* outperforms the second best MACLR in both settings of two datasets. Indirect supervision from DPR boosts performance against training from scratch (the second row from the bottom), while label thesaurus construction improves accuracy over those using textual label names (the third row from the bottom).

domain QA (Karpukhin et al., 2020) for reranking, where prediction scores on test set from *DDR* and Sentence-BERT are linearly combined as the final ranking score.

**Results** We report performance of *DDR* with(out) indirect supervision from dense retrieval for open-domain QA in Tab. 2. On average, *DDR* boosts performance on LF-Amazon-131K by 8.50% and LF-WikiSeeAlso-320K by 21.73% compared with the second best baseline MACLR. When only 1% labels are seen during training, we observe that *DDR* shows much better transferability than other pre-trained models, e.g., average performance improved by 12.59% over MACLR and 39.80% over MPNet. As more labels are available for fine-tuning the model, the performance gain from *DDR* over others becomes even more significant: 17.64% over MACLR and 43.38% over MPNet under 5% labels setting.

## 4.2 Ultra-fine Entity Typing

**Task** Entities can often be described by very fine grained-types (Choi et al., 2018) and the ultra-fine entity typing task aims at predicting one or more fine-grained words or phrases that describe the type(s) of that specific mention (Xu et al., 2022). Consider the sentence "He had blamed Israel for failing to implement its agreements." Besides *person* and *male*, the mention "He" has other very specific types that can be inferred from the context, such as *leader* or *official* for the "blamed" behavior and "implement its agreements" affair. Ultra-fine entity typing has a broad impact on various NLP tasks that depend on type understanding, including coreference resolution (Onoe and Durrett, 2020), event detection (Le and Nguyen, 2021) and relation extraction (Zhou and Chen, 2022).

**Datasets and Metrics** We leverage the UFET dataset (Choi et al., 2018) to evaluate benefits of *DDR* with indirect supervision from dense retrieval. Among 10,331 entity types, accurately selecting finer labels (121 labels such as *engineer*) was more challenging to predict than coarse-grained labels (9 labels such as *person*), and this issue is exacerbated when dealing with ultra-fine types (10,201 labels such as *flight engineer*). Following recent entity typing literature (Li et al., 2022; Du et al., 2022), we train *DDR* on (originally provided) limited

| Method | Precision | Recall | F1 |
|---|---|---|---|
| BiLSTM | 48.1 | 23.2 | 31.3 |
| LabelGCN | 50.3 | 29.2 | 36.9 |
| LDET | 51.5 | 33.0 | 40.1 |
| Box4Types | 52.8 | 38.8 | 44.8 |
| LRN | **54.5** | 38.9 | 45.4 |
| UniST | 50.2 | 49.6 | 49.9 |
| MLMET | 53.6 | 45.3 | 49.1 |
| LITE | 52.4 | 48.9 | 50.6 |
| Context-TE | 53.7 | 49.4 | 51.5 |
| *DDR* (w/o thesaurus) | 51.6 | 46.0 | 48.6 |
| *DDR* (scratch) | 53.0 | 50.5 | 51.7 |
| *DDR* (full) | 51.9 | **52.3** | **52.1** |

Table 3: Results of UFET task. *DDR* outperforms competitive baselines with (upper 5 methods) or without (lower 3 methods) inductive bias from task pre-training.

crowd-sourced examples without relying on distant resources such as knowledge bases or head words from the Gigaword corpus. We follow prior studies (Choi et al., 2018) to evaluate macro-averaged precision, recall and F1.

**Baselines** We consider two categories of competitive entity typing models as baselines: 1) methods capturing the example-label and label-label relations, e.g., BiLSTM (Choi et al., 2018) that concatenates the context representation learned by a bidirectional LSTM and the mention representation learned by a CNN, LabelGCN (Xiong et al., 2019) that learns to encode global label co-occurrence statistics and their word-level similarities, LRN (Liu et al., 2021a) that models the coarse-to-fine label dependency as causal chains, Box4Types (Onoe et al., 2021) that captures hierarchies of types as topological relations of boxes, and UniST (Huang et al., 2022a) that conduct name-based label ranking; 2) methods leveraging inductive bias from pre-trained models for entity typing, e.g., MLMET (Dai et al., 2021) that utilizes the pretrained BERT to predict the most probable words for "[MASK]" earlier incorporated around the mention as type labels, LITE (Li et al., 2022) and Context-TE (Du et al., 2022) that both leverage indirect supervision from pre-trained natural language inference.

**Implementation Details** The dataset UFET first asked crowd workers to annotate entity's types and then used WordNet (Miller, 1995) to expand these types automatically by generating all their synonyms and hypernyms based on the most common sense. Therefore, we automatically obtain label thesaurus entries from definitions and examples in WordNet and Wiktionary, which covers 99.99% of the whole label set. The time cost for prior label thesaurus construction is around 2.07 minutes.

Initialized with the original DPR checkpoint pre-trained on open-domain QA datasets, *DDR* firstly optimizes the model given positive example-label pairs and in-batch negatives. With the fine-tuned model, we then perform dense retrieval on the label set for each training example, keeping label documents with high scores but not in the true label set as hard negatives. *DDR* further updates the model with these additional hard negatives. For inference of mult-label entity type classification, we adopt labels with retrieval scores higher than a threshold that leads to the best F1 score on the development set.

**Results** In Tab. 3, we show performance of *DDR* and other entity typing methods. *DDR* obtains the state-of-the-art F1 score over the the best baseline training from scratch (LRN) by 6.7 and the best baseline with inductive bias from the language model (Context-TE) by 0.6.

### 4.3 Few-shot Intent Classification

**Task** As a fundamental element in task-oriented dialog systems, intent detection is normally conducted in the NLU component for identifying a user's intent given an utterance (Ham et al., 2020). Recently, accurately identifying intents in the few-shot setting has attracted much attention due to data scarcity issues resulted from the cost of data collection as well as privacy and ethical concerns (Lin et al., 2023b). Following the few-shot intent detection benchmark (Zhang et al., 2022), we focus on the challenging 5-shot and 10-shot settings.

**Datasets and Metrics** To evaluate the effectiveness of *DDR* for NLU with large decision spaces, we pick three challenging intent datasets with a relatively large number of semantically similar intent labels. Banking77 (Casanueva et al., 2020) is a single-domain dataset that provides very fine-grained 77 intents in a Banking domain. HWU64 (Liu et al., 2021b) is a multi-domain (21 domains) dataset recorded by a home assistant robot including 64 intents ranging from setting alarms, playing music, search, to movie recommendation. CLINC150 (Larson et al., 2019) prompts crowd workers to provide questions or commands in the manner they would interact with an artificially intelligent assistant covering 150 intent classes over 10 domains. We report accuracy

| Method | Banking77 | | HWU64 | | CLINC150 | |
|---|---|---|---|---|---|---|
| | 5-shot | 10-shot | 5-shot | 10-shot | 5-shot | 10-shot |
| *w/ Data Augmentation* | | | | | | |
| ICDA-$XS$ (+1x) | 80.29 | 86.72 | 81.32 | 85.59 | 91.16 | 93.71 |
| ICDA-$S$ (+4x) | 81.95 | 87.37 | 81.97 | 86.25 | 91.22 | 93.98 |
| ICDA-$M$ (+16x) | **84.01** | 88.64 | 81.84 | 87.36 | 91.93 | 94.71 |
| ICDA-$L$ (+64x) | 83.90 | 89.12 | 81.97 | 86.94 | 92.41 | 94.73 |
| ICDA-$XL$ (+128x) | 83.90 | **89.79** | **82.45** | **87.41** | **92.62** | **94.84** |
| *w/o Data Augmentation* | | | | | | |
| USE | 76.29 | 84.23 | 77.79 | 83.75 | 87.82 | 90.85 |
| ConveRT | 75.32 | 83.32 | 76.95 | 82.65 | 89.22 | 92.62 |
| USE+ConveRT | 77.75 | 85.19 | 80.01 | 85.83 | 90.49 | 93.26 |
| ConvBERT | – | 83.63 | – | 83.77 | – | 92.10 |
| + MLM | – | 83.99 | – | 84.52 | – | 92.75 |
| + MLM + Example | – | 84.09 | – | 83.44 | – | 92.35 |
| + Combined | – | 85.95 | – | 86.28 | – | 93.97 |
| DNNC | 80.40 | 86.71 | 80.46 | 84.72 | 91.02 | 93.76 |
| CPFT | 80.86 | 87.20 | 82.03 | **87.13** | 92.34 | 94.18 |
| - Pretraining | 76.75 | 84.83 | 76.02 | 82.96 | 88.19 | 91.55 |
| Context-TE | 80.76 | 85.53 | – | – | – | – |
| *DDR* (w/o thesaurus) | 78.86 | 84.16 | 80.30 | 83.74 | 88.86 | 91.58 |
| *DDR* (scratch) | 82.11 | 85.97 | 80.76 | 84.71 | 88.58 | 91.16 |
| *DDR* (full) | **83.86** ($XL$) | **88.25** ($M$) | **84.29** ($XL$) | 86.34 ($S$) | **92.71** ($XL$) | **94.58** ($M$) |

Table 4: Few-shot Intent Detection Accuracy on three benchmark datasets. *DDR* achieves much higher accuracy than existing strong learning baselines without data augmentation, while competitive with ICDA requiring a considerable amount of augmented data. ICDA prepares additional synthetic data with the scale ranging from the same amount (ICDA-$XS$) to 128 times (ICDA-$XL$) of the original few-shot train size. In the bottom row, we also mark performance of *DDR* with the most comparable ICDA variant. Results not available in original papers are marked as −.

for this single-label classification task.

**Baselines** There are two families of intent learning algorithms to cope with limited amounts of example-label pairs: 1) Classifiers based on sentence embeddings from PLMs, including USE (Yang et al., 2020) embeds sentences from 16 languages into a single semantic space using a multi-task trained dual-encoder, ConveRT (Henderson et al., 2020) that pretrains a dual-encoder model on Reddit Data by maximizing the similarity score of embeddings of positive input-response pairs; ConvBERT (Mehri et al., 2020) that adopts a BERT-base model pre-trained on a large open-domain dialogue corpus; 2) methods leverage semantic similarity between utterances from different users, e.g. ConvBERT+Combined (Mehri and Eric, 2021) that extends the ConvBERT model with task-adaptive masked language modelling (**MLM**) and infers the intent of the utterance based on the similarity to the **examples** corresponding to each intent, DNNC (Zhang et al., 2020a) that leverages BERT-style pairwise encoding to train a binary classifier that estimates the best matched training example for a user input, CPFT (Zhang et al., 2021) that conducts contrastive pre-training on example sentences

from six public intent datasets to discriminate semantically similar utterances.

Different from *DDR* and prior baselines, ICDA (Lin et al., 2023a), a most recent work tackles the challenge of limited data for intent detection by generating high-quality synthetic training data. It first fine-tunes a PLM on a small seed of training data for new data point synthesization, and then employs pointwise V-information (PVI) based filtering to remove unhelpful ones.[12]

**Implementation Details** Based on the observation that utterances labeled by identical intents share very similar sentence-level semantic meanings, we first utilize the whole set of unlabeled training examples to represent their corresponding pseudo labels predicted by Sentence-BERT (Reimers and Gurevych, 2019). We perform prediction on a single NVIDIA RTX A5000 GPU and it takes 4.17 minutes to complete label thesaurus construction for Banking77, 6.45 minutes for HWU64 and 7.61

---

[12]Although the learning and augmentation direction are orthogonal hence incommensurable, we display results with augmented training data as well to reflect the advantage of *DDR* over ICDA in terms of their required scale of augmented synthetic data to achieve similar good performance.

| Dataset | Setting | Sentence-BERT (1st round) | *DDR* (2nd round) |
|---------|---------|---------------------------|-------------------|
| Banking77 | 5-shot | **84.29** | 83.86 |
|           | 10-shot | 88.02 | **88.25** |
| HWU64 | 5-shot | 82.81 | **84.29** |
|       | 10-shot | 85.69 | **86.34** |
| CLINC150 | 5-shot | 91.82 | **92.71** |
|          | 10-shot | 94.07 | **94.58** |

Table 5: Impact on intent detection from label thesaurus constructed with predictions from Sentence-BERT and *DDR* for the unlabeled whole set of the training set.

minutes for CLINC150. After training *DDR* with this first version of label thesaurus, we then update pseudo labels for each training example with predictions from *DDR* for the second-round training.

Regardless of the basis for label thesaurus construction, *DDR* is trained for two phases in each round: only few-shot positive example-label pairs and in-batch negatives are used in the first phase, while labels wrongly predicted for training examples from the prior phase are used as additional hard negatives in the second phase. After the second round of training, the latest *DDR* makes predictions on the whole set of training examples for the final label thesaurus construction, from which *DDR* retrieves the label with the highest score as the prediction.

**Results** Tab. 4 presents performance of different intent detection methods on 5- and 10-shot setting. Without extra training data synthesization, we observe significant performance gain from the proposed *DDR* over existing baselines. Although ICDA obtains enhanced accuracy by increasing data augmentation scale, *DDR* is able to achieve comparable performance without crafting 4x even 128x synthetic instances. In Tab. 5, we additionally study the quality of constructed label thesaurus in accordance with the prediction on the unlabeled whole training set from Sentence-BERT and *DDR* after the first round of training. We find that with a more accurate pseudo connection between example and label leads to a higher quality of label thesaurus construction, and finally benefits intent detection.

## 5  Conclusions

In this paper, we focus on discriminative NLU tasks with large decision spaces. By reformulating these tasks as learning-to-retrieve tasks, we are able to leverage rich indirect supervision signals from dense retrieval. Moreover, by representing decision spaces with thesaurus, we provide rich semantic knowledge of each label to improve the understanding of labels for dense decision retrievers. Experiments on 6 benchmark datasets show the effectiveness of our method on decision spaces scaling from hundreds of candidates to hundreds of thousands of candidates. Future work can extend our method to more large-space decision making tasks, especially in the low-resource setting.

## Acknowledgement

We appreciate the reviewers for their insightful comments and suggestions. The logo for *DDR* used in this paper is sourced from the Wikipedia page[13] under CC BY-SA 3.0, for which we appreciate the contribution of the Wikipedia community.

Nan Xu is supported by the USC PhD Fellowship. Fei Wang is supported by the Annenberg Fellowship and the Amazon ML Fellowship. Mingtao Dong is supported by the Provost's Research Fellowship. Muhao Chen is supported by the NSF Grant IIS 2105329, the NSF Grant ITE 2333736, the DARPA MCS program under Contract No. N660011924033 with the United States Office Of Naval Research, a Cisco Research Award, two Amazon Research Awards, and a Keston Research Award. The computing of this work has been partly supported by a subaward of NSF Cloudbank 1925001 through UCSD.

## Limitations

The proposed *DDR* leverages the dual-encoder architecture with the inner dot product to compute embedding similarity, obtaining state-of-the-art performance on three investigated challenging large-space decision making tasks. More expressive models for embedding input and label thesaurus, such as joint encoding by a cross-encoder is not discussed. Moreover, other ways to model the connection between input and label thesaurus entry, such as using a sequence-to-sequence language model to generate the label name given input text and label thesaurus entry, is not explored yet. We believe adopting more advanced dense retrieval algorithms can further promote performance for large-space decision making. We leave this as an exciting future direction.

---

[13]https://en.wikipedia.org/wiki/File:Dance_Dance_Revolution_dance_pad_icon.png

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
