# OpenReview forum: "Dense Retrieval as Indirect Supervision for Large-space Decision Making"
_EMNLP/2023/Conference — EMNLP 2023 Findings_

### Official Review · Reviewer_FPkP · 2023-07-24

**Soundness:** 3

**Excitement:**

4: Strong: This paper deepens the understanding of some phenomenon or lowers the barriers to an existing research direction.

**Paper Topic And Main Contributions:**

The paper introduces a novel solution called Dense Decision Retrieval (DDR) for addressing the challenges of large label spaces in discriminative natural language understanding (NLU) tasks. The traditional approach of predicting fine-grained decisions as logits is replaced by a bi-encoder retrieval architecture that learns to retrieve decision thesaurus. Experimental results demonstrate that DDR outperforms baselines in various tasks like multi-label classification, entity typing, and intent classification.



**Questions For The Authors:**

- It seems that the proposed solution may face challenges in generalization when applied to scenarios outside the scope of the Wikipedia or WordNet collection. Can you offer further justification for this limitation?
- The relevance between the experimental tasks and "decision making" appears to be limited. Could you please provide additional justification for this connection?
- In sec 3.2, the LLM approach is mentioned as a potential method for thesaurus construction but is not well explored in the study. Could you please elaborate on the reasons for this decision?
- [minor] Given the impressive reasoning capabilities demonstrated by recent LLMs, do you believe that a generation-based framework could be a more promising solution compared to a retrieval-based framework? Why?


**Reasons To Accept:**

- The proposed task, large-space decision making, is important and may opens up opportunities for addressing more real-world challenges.
- The presented solution is clear and paper is well written
- Experiments are comprehensive


**Reasons To Reject:**

- The experimental tasks appear to lack direct relevance to decision making, and it seems that the construction of the thesaurus heavily relies on these specific tasks rather than being a general construction method applicable to various open-domain tasks.
- One of the main challenges in addressing large-space decision making lies in the vastness of the decision space, coupled with the lack of naturally-existing of ground truth explanations for each decision. However, the experiments seem to have been conducted with carefully designed tasks, where classification labels align well with existing knowledge bases like Wikipedia and WordNet. This controlled environment may not fully reflect the real-world decision-making scenarios as the authors mention in their motivation.
- Technical novelty and contribution is limited.

**Reproducibility:**

4: Could mostly reproduce the results, but there may be some variation because of sample variance or minor variations in their interpretation of the protocol or method.

**Reviewer Confidence:**

4: Quite sure. I tried to check the important points carefully. It's unlikely, though conceivable, that I missed something that should affect my ratings.

---

> ### Author Rebuttal · Authors · 2023-08-29
>
> We really appreciate your suggestions and would like to clarify some details as follows:
>
> **Relevance to "decision making"**
>
> From lines 194 to 205 in Section 3.1, we give a formal definition of the NLU tasks with large decision spaces, where ranking labels’ relevance to input text is considered as a decision making process. We understand that clarifying the difference from decision making in planning or reinforcement learning studies is necessary, and will do so at the beginning of the paper.
>
> **Generalizable construction process of label thesaurus**
>
> - We propose a **general solution** to address large label spaces of NLU tasks by reformulating it as a learning-to-retrieve task. It allows the flexibility of adding new labels from different task learning resources, and reusing shared ones among them.
>
> - Moreover, there is **no restriction** on strategies for label thesaurus construction, and we provide three possible ways from lines 217 to 230 in Section 3. We can rely on resources including definition/explanation for general concepts such as Wikipedia and WordNet. Besides, LLMs could be used to generate essential information for labels missing in existing lexicographical knowledge bases.
>
> **Investigated tasks with large label spaces**
>
> Our experiments covered various application areas: 1) product and document recommendation, 2) entity typing for type understanding, 3) intent classification  in task-oriented dialog systems. These tasks also covered  label space sizes ranging from hundreds, tens of thousands, to hundreds of thousands. In addition to these representative large-decision-space NLU tasks, we are happy to work on more evaluation on tasks of this kind upon the release of our implementation.
>
> **Technical contributions**
>
> In this paper, we present a novel and strong solution, DDR, for NLU tasks with large-space decision making that leverages indirect supervision from dense retrieval. We also provide semantically meaningful decision thesaurus construction that further improves the decision-making ability of DDR.
>
> **Scenarios outside the scope of the Wikipedia or WordNet collection**
>
> As mentioned from lines 221 to 230 in Section 3.2 Thesaurus Construction, we can refer to two other possible resources for label thesaurus construction: Large Language Models and Training examples. LLMs could be used to generate essential information for labels missing in existing lexicographical knowledge bases, while training examples could be used for thesaurus construction if instances with the same label share sentence-level semantic similarities.
>
>
> **Generation-based framework for NLU tasks with large label spaces**
>
> For now, LLMs still face critical, unresolved challenges to address NLU tasks with large label spaces. Specifically, a large number of label names cannot fit in the limited context of autoregressive LMs. Accurately matching next token probability to multiple tokens of label names is also non-trivial, and labels rarely seen in the pre-training corpus of LLMs are less likely to be generated. Hence, using PLMs and retrieval-based decision retrieval still represent a much more practical solution.

---

### Official Review · Reviewer_E5JK · 2023-08-01

**Typos Grammar Style And Presentation Improvements:** 030 - descriminative → discriminative…
**Soundness:** 3

**Excitement:**

3: Ambivalent: It has merits (e.g., it reports state-of-the-art results, the idea is nice), but there are key weaknesses (e.g., it describes incremental work), and it can significantly benefit from another round of revision. However, I won't object to accepting it if my co-reviewers champion it.

**Paper Topic And Main Contributions:**

The paper presents a novel approach called Dense Decision Retrieval (DDR) to address discriminative NLU tasks using dense retrieval. The authors argue that traditional classifier methods struggle with tasks involving a large number of labels or unseen labels due to data sparsity. To overcome this, they propose representing the label space with descriptive labels known as the 'label thesaurus' and employ the indirect supervision of DPR to retrieve label thesaurus entries. The results demonstrate that the DDR model consistently outperforms baselines on three tasks where the number of training labels is much smaller than the entire label space.

**Questions For The Authors:**

1. Could you please clarify the source used as the 'label thesaurus' in the intent classification task?
2. What does the 'w/o thesaurus' setup in Table 1, Table 2, and Table 3 refer to?
3. It is noted that large language models are mentioned as potential thesaurus sources, but they were not utilized in the experiments.


**Reasons To Accept:**

1. The paper introduces an innovative approach, Dense Decision Retrieval (DDR), to tackle discriminative NLU tasks using dense retrieval.
2. The proposed DDR model exhibits consistent improvements over baseline methods in the three tasks.


**Reasons To Reject:**

1. The authors may consider addressing some missing descriptions and typos that are affecting the readability of the paper.

**Reproducibility:**

4: Could mostly reproduce the results, but there may be some variation because of sample variance or minor variations in their interpretation of the protocol or method.

**Reviewer Confidence:**

4: Quite sure. I tried to check the important points carefully. It's unlikely, though conceivable, that I missed something that should affect my ratings.

---

> ### Author Rebuttal · Authors · 2023-08-29
>
> We thank your helpful suggestions and would like to clarify following details:
>
> **Label thesaurus construction for the intent classification task**
>
> - There are two rounds of label thesaurus construction as mentioned in lines 542-551.
>     - The first round: In 5- and 10-shot settings, we only have 5 or 10 samples per label, but abundant unlabeled examples. Therefore, we predict labels for examples and then concatenate example texts assigned with the same pseudo label to construct label thesaurus. Specifically, example labels are predicted according to their embedding similarity with the limited labeled samples based on Sentence-BERT.
>     - The second round: After training DDR with this first round of label thesaurus, we then update pseudo labels for each training example with predictions from DDR and start the second-round training with the new label thesaurus.
> - Normally, the second round of pseudo labels from DDP are more accurate than those from Sentence-BERT. We further show the benefits of this thesaurus construction process in Table 4, where label thesaurus after the second round leads to better classification performance than that after the first round.
>
> **'w/o thesaurus' setup in Table 1, Table 2, and Table 3**
>
> 'w/o thesaurus means no thesaurus construction but using only label names to represent the decision.
>
> **Utilization of Large Language Models in experiments for label  thesaurus construction**
>
> We use LLMs as a supplementary resource for thesaurus construction if necessary. For instance, we can find descriptions of 99.95% labels from LF-WikiSeeAlso-320K (line 358) from lexicographical knowledge bases. We then use the LLM (in this context, ChatGPT) to obtain relevant information for remaining 0.05% labels.

---

### Official Review · Reviewer_YGUm · 2023-08-05

**Soundness:** 3

**Excitement:**

3: Ambivalent: It has merits (e.g., it reports state-of-the-art results, the idea is nice), but there are key weaknesses (e.g., it describes incremental work), and it can significantly benefit from another round of revision. However, I won't object to accepting it if my co-reviewers champion it.

**Paper Topic And Main Contributions:**

This paper proposes a retrieval-based method for classification. The method is motivated by the issue of rare labels in long-tailed label distributions and generalization to unseen labels.

The frames decision-making as a retrieval over a thesaurus (as well as corpus of training data), and shows performance gains in limited-label scenarios.

**Questions For The Authors:**

What is the computational cost for constructing the thesaurus in each scenario?

Can the thesaurus be expanded at any point?

To accomodate more tasks, presumably the thesaurus would have to grow continually along with the growing label space. Can the proposed thesaurus scale in such scenarios?

**Reasons To Accept:**

The issue of rare labels is important, and the proposed model has good performance in limited-label scenarios.

The construction of a thesaurus based on external sources including lexical knowledge bases and large language models is interesting and shown to be effective.

**Reasons To Reject:**

LF-Amazon-131K and LF-WikiSeeAlso-320K results are reported for 1% and 5% labels, but this two settings alone are not entirely convincing.

The model depends heavily on the thesaurus, but ablation or detailed discussion of thesaurus construction and its impact on performance is lacking. Besides, thesaurus must be constructed for each dataset, such as Wikipedia or Amazon products data.

Overall, I do not think the method has significant novelty over standard dense retrieval methods or retrieval-augmented decision making models.

**Reproducibility:**

4: Could mostly reproduce the results, but there may be some variation because of sample variance or minor variations in their interpretation of the protocol or method.

**Reviewer Confidence:**

3: Pretty sure, but there's a chance I missed something. Although I have a good feel for this area in general, I did not carefully check the paper's details, e.g., the math, experimental design, or novelty.

---

> ### Author Rebuttal · Authors · 2023-08-29
>
> We sincerely appreciate your constructive feedback and would like to clarify some concerns as follows:
>
> **1% and 5% setting of Extreme Multi-label Classification task on LF-Amazon-131K and LF-WikiSeeAlso-320K**
> - One of our goals is to demonstrate the effectiveness of indirect supervision in **low-resource settings**. We therefore follow the low-shot setting introduced in [1] where the training subset covers 1%/5% labels from the whole set, instead of using a full-shot setting where training data are enough.
> - To comprehensively evaluate benefits of dense retrieval under different scenarios of large decision spaces with limited per-label training data, we also consider 1) thousands of labels with different granularities for entity typing in Section 4.2, and 2) few-shot setting on intent classification in Section 4.3.
>
> [1] Xiong, Yuanhao, et al. "Extreme zero-shot learning for extreme text classification." arXiv preprint arXiv:2112.08652 (2021).
>
> **Details about thesaurus construction**
>
> We describe thesaurus construction details in Implementation Details paragraph for three tasks from lines 354 to 365 in Section 4.1, from lines 452 to 460 in Section 4.2 and from lines 542 to 551 in Section 4.3. We will also release the code for thesaurus construction soon.
>
> **Construction cost  of label thesaurus**
>
> We perform label thesaurus construction on Intel(R) Xeon(R) Gold 5217 CPU @ 3.00GHz with 32 CPUs and 8 cores per CPU. For labels thesaurus constructed based on training samples in few-shot intent classification (last three datasets), we predict pseudo labels with Sentence-BERT on a single NVIDIA RTX A5000 GPU. The concrete time cost per dataset is as follows:
> - LF-WikiSeeAlso-320K: 59.12 min
> - LF-Amazon-131K: 28.24 min
> - UFET: 2.07 min
> - Banking77: 4.17 min
> - HWU64: 6.45 min
> - CLINC150: 7.61min
>
> **Expansion (scalability) of label thesaurus**
>
> Yes, the thesaurus is freely expandable. As discussed from lines 54 to 63 in Introduction, simply training a classifier is not adaptable to unseen labels. The proposed DDR, however, can expand thesaurus by adding the new label as a new entry in the same approach to initial (automated) construction.
>
> The design of DDR is sufficiently flexible to support continual learning as the label space grows in real-world applications: 1) fine-tuning DDR when new instances and labels are added, and 2) encoding new label texts into thesaurus with fast inference supported by FAISS.
>
> **Ablation study of label  thesaurus**
>
> For all three investigated tasks with large decision spaces, we’ve reported results of not using the thesaurus but only textual label names as shown in rows DDR (w/o thesaurus) in Table 1, 2 and 3. We observe that the proposed DDR (full) benefits greatly from the constructed thesaurus, obtaining much higher accuracy than DDR (w/o thesaurus).
>
> **Thesaurus construction wrt. Dataset**
>
> Traditional classifiers for NLU tasks with large decision spaces have to be trained separately for different datasets or decision spaces. However, our label thesaurus is based on the concrete label space and then constructed automatically according to publically available resources. It could be reused for different tasks/datasets with label overlapping, e.g., training subset covers 1% and 5% labels for extreme multi-label classification tasks discussed in Section 4.1. It is also freely expandable.
>
>
> **Novelty over standard dense retrieval methods or retrieval-augmented decision making models**
>
> - Difference from standard dense retrieval methods: this paper investigates how to leverage indirect supervision from dense passage retrieval to address NLU with large label spaces. Specifically, we provide semantically meaningful decision thesaurus construction that further improves the decision-making ability of DDR.
> - Difference from retrieval-augmented decision making models: retrieval-augmented approaches seek to retrieve auxiliary knowledge from external resources to augment the contextual information of an autoregresive LM, for which we do not see obvious connection to our work that adapts pretrained bi-encoders for decision retrieval.

---

### Meta-Review · Area_Chair_BY7t · 2023-09-18

**Recommendation:** 4

**Metareview:**

This paper utilized dense retrieval to address discriminative NLU tasks. The proposed task is well motivated and may yield some more real-world impact. The paper is well written in general. Reviewers praised the comprehensive experiments with consistent improvements of the proposed model over baselines. On the other hand, two reviewers raised concerns about limited technical novelty of the proposed method given the existing work. The readability of the paper can also be improved. Overall, I think this paper proposes a solid approach (while not super innovative) to a well-motivated task with convincing experiments. I hope the reviewers' comments on the weaknesses will help improve the quality of the next version of the paper.

---

### Decision · Program_Chairs · 2023-10-07

**Decision:**

Accept-Findings

**Comment:**

This paper utilized dense retrieval to address discriminative NLU tasks. The proposed task is well motivated and may yield some more real-world impact. The paper is well written in general. Reviewers praised the comprehensive experiments with consistent improvements of the proposed model over baselines. On the other hand, two reviewers raised concerns about limited technical novelty of the proposed method given the existing work. The readability of the paper can also be improved. Overall, I think this paper proposes a solid approach (while not super innovative) to a well-motivated task with convincing experiments. I hope the reviewers' comments on the weaknesses will help improve the quality of the next version of the paper.